# GatorTron and GatorTronGPT: Large Language Models for Clinical Narratives

**Cheng Peng[1], Xi Yang[1,2], Mengxian Lyu[1], Kaleb E Smith[3], Anthony B. Costa[3], Mona G. Flores[3], Jiang Bian[1,2], Yonghui Wu[1,2]**

[1]Department of Health Outcomes and Biomedical Informatics, College of Medicine, University of Florida
[2]Cancer Informatics Shared Resource, University of Florida Health Cancer Center
[3]NVIDIA, Santa Clara, California, USA
{c.peng, alexgre, lvmengxian, bianjiang, yonghui.wu}@ufl.edu, {kasmith, mflores, acosta}@nvidia.com

## Abstract

Large language models (LLMs) have become the foundational technology for natural language processing (NLP). We introduce clinical LLMs including GatorTron and GatorTronGPT, summarize their applications, highlight the impact on clinical NLP and artificial intelligence (AI) applications, and provide insights in using LLMs for medical AI applications.

## Introduction

Large language models (LLMs) have become the foundational technology used to explore clinical narratives for artificial intelligence (AI) applications in the medical do-main. There is a surge(Singhal et al. 2023a) of studies exploring LLMs for various medical research as the success of ChatGPT. For example, Galactica(Taylor et al. 2022) was trained using a large corpus of papers, reference material, and knowledge bases, which demonstrated good performance on benchmark datasets including PubMedQA(Jin et al. 2019) and MedMCQA(Pal, Umapathi, and Sankarasubbu 2022). Med-PaLM(Singhal et al. 2023a) and later Med-PaLM 2(Singhal et al. 2023b) achieved an "expert" level of performance on the MedQA dataset of USMLE2-style questions. However, most existing studies focus on general-purpose LLMs such as ChatGPT. There are studies developing LLMs using biomedical literature or fine-tuning general-purpose LLMs using a small set of clinical notes from the MIMIC data-base. Nevertheless, there are limited studies developing and examining clinical LLMs developed using clinical text data. We previously have developed clinical LLMs including GatorTron and GatorTronGPT, which have been widely used in the clinical domain. This abstract introduces our clinical LLMs and their applications in the clinical do-main, provides resources to facilitate the use of GatorTron models, and highlights their impact on clinical NLP.

## GatorTron and GatorTronGPT Models

GatorTron(Yang et al. 2022) was trained from scratch using 90 billion words of text from the de-identified clinical notes of the University of Florida Health (82 billion

words), MIMIC-III corpus (0.5 billion words), PubMed articles (6 billion words), and Wikipedia (2.5 billion words). We adopted the BERT architecture and developed three different configurations including 345 million parameters (i.e., GatorTron-base), 3.9 billion parameters (i.e., GatorTron-medium), and 8.9 billion parameters (i.e., GatorTron-large). GatorTron models are encoder-only LLMs, where only the encoder component of the transformer was used. GatorTron models are available from: https://huggingface.co/UFNLP.

GatorTronGPT(Peng et al. 2023a) was trained from scratch using 277 billion words from the de-identified clinical text of UF Health (82 billion words) and 195 billion words of diverse English text from the Pile(Gao et al. 2020) dataset. GatorTronGPT adopted the GPT-3 architecture with 5 and 20 billion parameters. GatorTronGPT is a decoder-only generative clinical LLM, where only the decoder of the transformer was used.

## GatorTron and GatorTronGPT Applications

GatorTron has been applied to many clinical NLP tasks including clinical concept extraction, medical relation extraction (RE), Semantic Textual Similarity (STS), medical Natural Language Inference (NLI), and medical question answering (QA). GatorTron achieved state-of-the-art performance on many benchmark datasets including i2b2 2010(Uzuner et al. 2011), i2b2 2012(Sun, Rumshisky, and Uzuner 2013), and n2c2 2018(Henry et al. 2020). In addition, Table 1 summarizes the clinical applications of GatorTron models in other studies. GatorTron has been applied for predictions of readmission, mortality, length of stay, insurance denial, disease response, and mobility function. Though many studies focus on the generative LLMs based on the decoder-only architecture, the encoder-only LLM, GatorTron, has been widely used for many medical AI applications

As a generative LLM adopted the decoder-only architecture, GatorTronGPT solved many clinical NLP using unified text-to-text learning, including clinical concept extraction, concept normalization, relation extraction, abbreviation disambiguation, NLI, attribute filling, progress notes understanding. GatorTronGPT was applied to generate narrative sections of clinical notes, which physicians can not differentiate them from real-world clinical notes. Our recent study(Peng et al. 2023a) proposed a unified text-to-

| | Model | Evaluation Task | Evaluation Dataset | Performance Ranking |
|---|---|---|---|---|
| Peng et al. (Peng et al. 2023c) | GatorTron-base | Clinical concept extraction End-to-end clinical relation extraction | 2018 n2c2 dataset 2022 n2c2 dataset 2018 n2c2 dataset 2022 n2c2 dataset | 1/14 1/7 1/7 2/7 |
| Jiang et al. (Jiang et al. 2023) | GatorTron-base | Readmission In-hospital mortality prediction Comorbidity index prediction Length of stay (LOS) prediction Insurance denial prediction | - | 1/6 (Average prediction on five tasks) |
| Tan et al. (Tan et al. 2023) | GatorTron-base | Natural Language Inference (infer cancer disease response from radiology reports) | RECIST dataset | 1/14 |
| Chen et al. (Chen et al. 2023b) | GatorTron-base | Medication mention extraction Event classification Context classification | 2022 n2c2 dataset | 3/6 2/6 1/6 |
| Pathak et al. (Pathak et al. 2023) | GatorTron-base | Clinical concept extraction (thyroid nodule characteristics extraction) | Ultrasound reports from UF Health | 1/5 |
| Chen et al. (Chen et al. 2023a) | GatorTron-base | Clinical concept extraction (delirium symptom extraction) | Delirium symptoms corpus from UF Health | 1/8 |
| Ong et al. (Ong et al. 2023) | GatorTron-base | Disease response classification | CT and MRI reports | 1/4 |
| Alameldin et al.(Alameldin and Williamson 2023) | GatorTron-base | Natural Language Inference Evidence retrieval | SemEval 2023-Task 7 | 1/6 1/4 |
| Le et al. (Gao et al. 2020) | GatorTron-base | Mobility functioning classification | n2c2 clinical notes | 3/9 |
| Ge et al. (Ge et al. 2023) | GatorTron-base | Multi-label text classification (medical diagnosis prediction) | RAA dataset MIMIC-III | 1/9 1/9 |
| Peng et al. (Peng et al. 2023b) | GatorTron-base GatorTron-medium GatorTron-large | Clinical concept extraction End-to-end clinical relation extraction | 2018 n2c2 dataset 2022 n2c2 dataset 2018 n2c2 dataset 2022 n2c2 dataset | 1/5 1/5 1/5 1/3 |

Table 1: Performance ranking of GatorTron-base models in various evaluation tasks.

text learning architecture, which solved seven major clinical NLP tasks using a unified GatorTronGPT model using a strategy to freeze LLMs, i.e., keep the model parameters unchanged during prompting.

models to fill the gap of sharing clinical corpora. In addition, GatorTronGPT provides a solution to solve many diverse information extraction and classification tasks using unified text-to-text learning.

## Conclusion and Discussion

GatorTron and GatorTronGPT models have greatly improved clinical NLP and medical AI applications. The encoder-only GatorTron models have been widely used for patient information extraction and various prediction-based tasks. The decoder-only GatorTronGPT can generate synthetic clinical text for the development of synthetic NLP

Most evaluations of LLMs used standard NLP tasks and datasets, there is an absence of holistic evaluation frameworks to examine LLMs in real-world health, which is a significant disconnection between the LLM evaluation regimes and expected clinical benefits(Wornow et al. 2023).

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
