# OpenReview forum: "GatorTron and GatorTronGPT: Large Language Models for Clinical Narratives"
_AAAI.org/2024/Spring_Symposium_Series/Clinical_FMs — AAAI 2024 SSS on Clinical FMs_

### Official Review · Reviewer_eN2v · 2024-02-16

**Rating:** 7
**Confidence:** 4

**Review:**

Quality and clarity:
- The writing is generally clear and easy to understand

Originality and significance:
- Gatertron Foundational Model trained on clinical data, highly relevant for this workshop

Weaknesses
- Performance ranking makes it more difficult to visualize the actual performance of the model, more fine-grained evaluation metrics would be appreciated, like accuracy for QA, precision and recall for the extraction tasks, etc
- Where are the GatertronGPT evaluation results?

---

### Official Review · Reviewer_NU4V · 2024-02-17
**GatorTron**

**Rating:** 5
**Confidence:** 4

**Review:**

This paper provides an overview of GatorTron's use in existing literature. While the authors raise concerns about existing LLMs for medical applications, the paper does not delineate the methodological novelty and performance improvement of the presented model. A crucial ambiguity lies in the selection process of the cited studies, leaving readers uncertain whether it constitutes a systematic review of evidence. Overall, the paper offers some use cases of GatorTron within the literature.

---

### Official Review · Reviewer_1meb · 2024-02-19
**Summary paper of previous excellent work in the development and use of the GatorTron family of models for clinical NLP tasks.**

**Rating:** 8
**Confidence:** 3

**Review:**

1. Summary and contributions: Briefly summarize the paper and its contributions
- The paper outlines the development of billion parameter GatorTron, an encoder only, and GatorTronGPT, a decoder only, transformers.
- They are trained from scratch on billions of text tokens, both from general text corpora as well as de-identified clinical notes. The paper then goes on to summarise the previously published performance of the models on commonly used clinical NLP benchmark tasks.

2. Strengths: Describe the strengths of the work. Typical criteria include: soundness of the claims (theoretical grounding, empirical evaluation), significance and novelty of the contribution, and relevance.
- It is a huge (maybe unique?) technical achievement for an academic institution to be able to train billion parameter LLMs on billions of tokens.
- The performance on various benchmarks is very strong.
- Highly relevant to this symposium

3. Weaknesses: Explain the limitations of this work along the same axes as above.
- The hypothesis of this work is that explicitly including clinical notes in an LLM’s training data aids performance on downstream clinical tasks. Therefore, a direct comparison to state-of-the-art general LLMs (e.g. GPT/Gemini/Llama/Mistral) in Table 1 would strengthen this argument. Especially since general LLMs have been through RLHF, unlike GatorTronGPT
- Not clear why the models were trained from scratch instead of finetuning a generally pre-trained model. Although the training corpus size is large, it is still dramatically smaller than that used in open-source models (such as Llama 2 trillion tokens)
- Does GatorTronGPT exhibit zero/few-shot learning abilities, or must it be finetuned?
- No section on ethical considerations or the ethics board approval. This is highly relevant given the large-scale use of (de-id) clinical notes and the subsequent open-sourcing of GatorTron. This is a great initiative, so it would be helpful to be explicit on how this was achieved so others could do the same in the future.
4. Correctness: Are the claims and method correct? Is the empirical methodology correct?
- “GatorTronGPT provides a solution to solve many diverse information extraction and classification tasks using unified text-to-text learning” seems like a strong statement.
- The methodology, although only briefly outlined due to page limit, is sound.
5. Clarity: Is the paper well written?
- The paper is well written. However, the last paragraph of the conclusion is oddly formatted and does not seem to follow from the previous one.
- The repetition of Peng et al b and c on seemingly the same datasets and tasks but with differing performance rankings in Table 1 is a little confusing.
- Would be useful to note the context window of the models.
6. Relation to prior work: Is it clearly discussed how this work differs from previous contributions?
- Yes, a clear explanation of general-purpose LLMs and their evaluation in biomedical/clinical NLP space. This then goes on to outline the limited specialised training of LLMs in the clinical domain. Although the motivation for this is implicit rather than explicit (training on clinical data may lead to better results on clinical tasks)
7. Reproducibility: Are there enough details to reproduce the major results of this work?
- No, but that is out of scope, given the page limit. The author could add the compute required to train the models.
- Links to the open source model weights or training code would be helpful.

---

### Official Review · Reviewer_c1UJ · 2024-02-21
**This is a clear review of two significant clinical LLMs though lack of originality as a review.**

**Rating:** 6
**Confidence:** 3

**Review:**

The abstract introduces two clinical LLMs, GatorTron and GatorTronGPT, and their applications in a clinical context. To this end, the authors review the training dataset of the two LLMs and summarize their performances in various evaluation tasks within a medical context. The abstract highlights the necessity of evaluating LLMs specialized for clinical purposes. The detailed description of the applications of the two LLMs provided a comprehensive picture for audience. Some concerns arose when I was reading the abstract:
1. The abstract tries to provide resources to facilitate the use of GatorTron, but I could hardly find lines addressing this claim throughout the abstract.
2. GatorTron and GatorTronGPT were distinguished from general-purpose LLMs in that they were trained with medical corpora. Hence, I’m curious about the performance contrast of a general-purpose model like ChatGPT (as a benchmark), and a specialized medical LLM like GatorTron. A direct comparison such as accuracy scores in medical NLP tasks (if it is possible to quantify their performance) between the two kinds of model will do.
3. The “Conclusion and Discussion” section gives a nice summary about the applications of the two LLMs in clinical contexts. From my personal conjecture, readers may want to know more about the state-of-art news of GatorTron and GatorTronGPT, such as the technical challenges we are dealing with. Put such questions in a bigger picture, audience may be interested in where we are going to, and what we will be able to do with GatorTron and GatorTronGPT in the future. I believe these topics fit well in the discussion session and can contribute to the discussion in the symposium.
In sum, the abstract provides a clear review of GatorTron and GatorTronGPT, a significant work in clinical NLP. As a review of existing studies, this abstract is inevitably short of originality. An additional discussion on the state-of-art and limitations of the present models will make the abstract more beneficial to the conference. Recommended.